# Water, sanitation, handwashing, and nutritional interventions can reduce child antibiotic use: evidence from Bangladesh and Kenya

Ayse Ercumen [1] ✉, Andrew N. Mertens [2], Zachary Butzin-Dozier[2], Da Kyung Jung [2], Shahjahan Ali [3], Beryl S. Achando[4], Gouthami Rao[5], Caitlin Hemlock[2], Amy J. Pickering [6,7], Christine P. Stewart [8] Sophia T. Tan[9], Jessica A. Grembi [9], Jade Benjamin-Chung [7,10], Marlene Wolfe[11], Gene G. Ho[2], Md. Ziaur Rahman[3], Charles D. Arnold[8], Holly N. Dentz[8], Sammy M. Njenga[12], Theodora Meerkerk[4], Belinda Chen[2], Maya Nadimpalli[11], Mohammad Aminul Islam [13], Alan E. Hubbard [2], Clair Null[14], Leanne Unicomb[3], Mahbubur Rahman [3,15], John M. Colford Jr[2], Stephen P. Luby [9], Benjamin F. Arnold [16] & Audrie Lin [17]

Antibiotics can trigger antimicrobial resistance and microbiome alterations. Reducing pathogen exposure and undernutrition can reduce infections and antibiotic use. We assess effects of water, sanitation, handwashing (WSH) and nutrition interventions on caregiver-reported antibiotic use in Bangladesh and Kenya, longitudinally measured at three timepoints among birth cohorts (ages 3–28 months) in a cluster-randomized trial. Over 50% of children used antibiotics at least once in the 90 days preceding data collection. In Bangladesh, the prevalence of antibiotic use was 10–14% lower in groups receiving WSH (prevalence ratio [PR] = 0.90 (0.82–0.99)), nutrition (PR = 0.86 (0.78–0.94)), and nutrition+WSH (PR = 0.86 (0.79–0.93)) interventions. The prevalence of using antibiotics multiple times was 26–35% lower in intervention arms. Reductions were largest when the birth cohort was younger. In Kenya, interventions did not affect antibiotic use. In this work, we show that improving WSH and nutrition can reduce antibiotic use. Studies should assess whether such reductions translate to reduced antimicrobial resistance.

Antimicrobial resistance (AMR) was associated with an estimated 4.95 million deaths in 2019[1], most in low-and-middle income countries (LMICs). Community carriage of antimicrobial resistant bacteria[2] and their abundance in sewage[3] is higher in LMICs—especially in Sub-Saharan Africa and South Asia—than high-income countries. Reasons may include densely populated conditions, lack of safe drinking water and sanitation[4], and widespread availability and frequent use of

antibiotics for both humans and animals[5,6]. A longitudinal study following birth cohorts in eight countries in South America, sub-Saharan Africa and Asia found the highest pediatric antibiotic use in South Asia[7]. In Bangladesh, 98% of the children in the study used antibiotics before the age of 6 months and had an average of 10 courses of antibiotics per child-year[7]. Similarly, in Kenya, children have an average of 22 antibiotic prescriptions between birth and age of 5 years[8]. In

comparison, a child in the US takes <2 courses of antibiotics per child-year[9]. Antibiotics are commonly prescribed in LMICs for diarrheal and respiratory infections[7,8]. While cultural factors such as the beliefs and perceptions of both the prescribers and consumers can drive antibiotic use[10], reducing the occurrence of diarrheal and respiratory infections may lead to reduced antibiotic use in LMICs.

Children are susceptible to enteric and respiratory infections when they are frequently exposed to pathogens due to poor water, sanitation, and hygiene (WASH) conditions. As of 2020, 30% of sub-Saharan Africa (and 13% of rural sub-Saharan Africa) had access to safely managed water services[11]. Similarly, 21% of sub-Saharan Africa and 47% of South Asia had access to safely managed sanitation in 2020[11]. Poor WASH conditions are estimated to account for 62% of deaths from diarrhea among children <5 years globally[12]. Early life antibiotic use (e.g., to treat childhood diarrheal infections) can further increase the risk of diarrhea[13,14]. Repeated episodes of diarrhea can lead to malnutrition, and malnourished children in turn experience increased incidence, duration and severity of diarrhea ("infection-malnutrition cycle")[15]. Improving WASH conditions in LMICs may reduce antibiotic use by decreasing incidences of childhood diarrhea[16] and respiratory infections[17]. Similarly, improved nutrition can make children less susceptible to infections[18] and reduce subsequent antibiotic use. A modeling study estimated that universal access to improved water and sanitation could reduce antibiotic use by 60%[19]. However, there are scarce empirical data on the effect of WASH and nutrition interventions on antibiotic use by children. A single randomized controlled trial to date found that community-level water chlorination reduced reported antibiotic use by 7% among children in urban Bangladesh, along with a 23% reduction in diarrhea[20].

Here, we utilize data from two cluster-randomized controlled trials of individual and combined water, sanitation, handwashing (WSH) and nutrition interventions in rural Bangladesh and Kenya. In Bangladesh, children receiving WSH and nutrition interventions had reduced prevalence of diarrhea[21] and respiratory infections[22] compared to controls. In Kenya, children receiving the nutrition intervention had marginally lower prevalence of respiratory infections than controls; there were no other intervention effects[23,24]. In both countries, the nutrition intervention improved child linear growth[21,23]. It is plausible that children that experienced fewer infections or better growth consequently used less antibiotics during the trial. The objective of the current study was to assess whether these interventions reduced caregiver-reported antibiotic use in children in rural Bangladesh and Kenya. For each country, we assessed the effects of the WSH, nutrition and nutrition plus WSH (N + WSH) interventions compared to controls, and the effects of the combined N + WSH intervention compared to WSH and nutrition interventions alone, with subgroup analyses by child age and sex. We followed CONSORT guidelines for cluster-randomized trials (Supplementary Table 1). Our findings demonstrate that, in Bangladesh, all three interventions reduced caregiver-reported antibiotic use, combining nutrition and WSH interventions did not lead to additional reductions, and the reductions were largest when the birth cohort was youngest. In contrast, the interventions did not reduce caregiver-reported antibiotic use in Kenya.

## Results
### Enrollment
In Bangladesh, 5551 pregnant women in 720 clusters were enrolled between 31 May 2012 and 7 July 2013. Caregiver-reported antibiotic use was recorded among children in the birth cohort participating in a longitudinal substudy conducted to assess environmental enteric dysfunction (EED), which included 1131 children at mean age 3 months (interquartile range [IQR] = 1.6–4.1 months), 1531 children at 14 months (IQR = 12.8–15.5 months) and 1531 children at 28 months (IQR = 27.2–29.7 months) (Supplementary Fig. 1). Of these, antibiotic

data were available for 1102 children (97%) at 3 months and 1528 children (>99%) each at 14 and 28 months (Supplementary Fig. 1).

In Kenya, 8246 pregnant women in 702 clusters were enrolled between 27 November 2012 and 21 May 2014. The EED substudy included 1493 children at mean age 6 months (IQR = 4.1–6.8 months), 1504 children at 17 months (IQR = 15.3–18.3 months) and 1444 children at 22 months (IQR = 21.1–23.7 months) (Supplementary Fig. 2). Antibiotic data were available for 1438 children (96%) at 6 months, 1449 children (96%) at 17 months and 1393 children (97%) at 22 months (Supplementary Fig. 2).

Among households enrolled in the EED substudy, characteristics were balanced across study arms and similar to the households enrolled in the full trial for both countries (Supplementary Tables 2, 3). Children lost to follow-up at the latter two measurement points were similar in their characteristics to those that completed all three follow-ups (Supplementary Tables 4, 5). In Bangladesh, among the subset of children enrolled in the EED study, those who received WSH and N + WSH interventions had 21–35% lower prevalence of diarrhea and those who received N + WSH interventions had 21-44% lower prevalence of acute respiratory infections, compared to controls (Supplementary Table 6). Notably, at age 3 months, all three interventions reduced the prevalence of acute respiratory infections by 28-44% (Supplementary Table 6). In Kenya, there were no intervention effects on diarrhea and respiratory infections among the subset of children enrolled in the EED substudy (Supplementary Table 7).

### Antibiotic use
In Bangladesh, 63% (635) of children in the control group used antibiotics at least once and 25% (248) multiple times in the last 90 days, for a mean of 5 total days (standard deviation [SD] = 6) (Table 1). The prevalence of using antibiotics at least once in the last 90 days in the control group was highest (75%) at the 14-month measurement, and similar for boys (64%) and girls (62%) (Supplementary Table 8). Caregivers reported 24 distinct antibiotics. The most common antibiotic classes were penicillins (34.4%), cephalosporins (30.7%) and macrolides (22.6%), while the most common antibiotics were amoxycillin (32.1%), azithromycin (19.6%) and cefixime (12.3%).

In Kenya, 53% (601) of children in the control group used antibiotics at least once and 13% (152) multiple times in the last 90 days, for a mean of 3 total days (SD = 4) (Table 1). Use appeared highest at the 6-month measurement and similar for boys (52%) and girls (53%) Supplementary Table 9. Caregivers reported 15 distinct antibiotics. The most common classes were sulfonamides (52.6%), penicillins (39.6%) and nitroimidazoles (7.2%), and the most common antibiotics were cotrimoxazole (52.6%), amoxycillin (38.7) and metronidazole (7.2%).

### Intervention effects on antibiotic use
All interventions reduced antibiotic use in Bangladesh compared to controls (Fig. 1, Fig. 2). The percentage of children who used antibiotics at least once in the last 90 days was 10–14% lower among children receiving any intervention than controls (WSH prevalence ratio [PR] = 0.90, 95% CI: 0.82–0.99, $p = 0.03$; nutrition PR = 0.86, 95% CI: 0.78–0.94, $p < 0.001$, N + WSH PR = 0.86, 95% CI: 0.79-0.93, $p < 0.001$, Figs. 1, Supplementary Table 10). The percent of children who used antibiotics multiple times in the last 90 days was 26-35% lower among children receiving interventions than controls (WSH PR = 0.74, 95% CI: 0.63-0.87, $p < 0.001$, nutrition PR = 0.66, 95% CI: 0.56–0.79, $p < 0.001$; N + WSH PR = 0.65, 95% CI: 0.55-0.78, $p < 0.001$, Fig. 1, Supplementary Table 10). In all intervention arms, episodes of antibiotic use was reduced by 0.17–0.21 episodes and total days of antibiotic use by approximately 1 day compared to controls (Fig. 2, Supplementary Table 10). The N + WSH intervention did not additionally reduce antibiotic use compared to the WSH and nutrition interventions (Fig. 1, Fig. 2, Supplementary Table 11). In Kenya, interventions were not associated with reduced antibiotic use compared to controls (Fig. 1,

Fig. 2, Supplementary Table 12). In the N + WSH arm, antibiotic use was similar to the WSH and nutrition arms (Fig. 1, Fig. 2, Supplementary Table 13).

## Effect modification

In Bangladesh, most antibiotic use metrics showed effect modification by child age (interaction *p*-values <0.20, Supplementary Tables 14, 15), and intervention effects were strongest for the youngest children. Children at age 3 months in any intervention arm experienced reduced antibiotic use compared to controls (Fig. 3, Fig. 4). In this age group, the percent of children who used antibiotics at least once in 90 days was 24–33% lower in the nutrition and N + WSH arms (Supplementary Tables 14, 15). Notably, in this age group, the percent of children who used antibiotics multiple times was 43% lower in the WSH arm (PR = 0.57, 95% CI: 0.37–0.86), 49% lower in the nutrition arm (PR = 0.51, 95% CI: 0.33–0.78) and 52% lower in the N + WSH arm (PR = 0.48, 95% CI: 0.33–0.70) compared to controls (Supplementary Tables 14, 15). In this age group, all interventions also reduced episodes of antibiotic use by 0.25–0.34 episodes and total days of antibiotic use by 1.23-1.52 days (Supplementary Table 15). At 14 months, interventions reduced the percent of children who used antibiotics multiple times by 23-30% and

episodes of antibiotic use by 0.15–0.19 episodes (Supplementary Tables 14, 15). Days of antibiotic use was reduced by approximately 1 day in the WSH and nutrition arms but not N + WSH arm compared to controls (Supplementary Table 15). At 28 months, the WSH intervention had no effect on any metric of antibiotic use (Fig. 3, Fig. 4). The nutrition intervention reduced single antibiotic use by 15% and episodes of antibiotic use by 0.12 episodes, while the N + WSH intervention reduced multiple antibiotic use by 36%, episodes of antibiotic use by 0.15 episodes and days of antibiotic use by approximately 1 day (Supplementary Tables 14, 15).

In Bangladesh, use of antibiotics at least once in 90 days showed effect modification by child sex (interaction *p*-values <0.20, Supplementary Tables 14, 15); all three interventions reduced the percent of girls but not boys who used antibiotics at least once in 90 days (Fig. 5, Fig. 6). All interventions reduced the percent children who used antibiotics multiple times in 90 days and the episodes and days of antibiotic use similarly for both sexes (interaction *p*-values > 0.20, Fig. 5, Fig. 6, Supplementary Tables 14, 15). Overall, the N + WSH intervention was not any more effective than the WSH and nutrition interventions in subgroup analyses by age and sex (Figs. 3–6, Supplementary Tables 14, 15).

In Kenya, there was no evidence of effect modification by child age or sex for most comparisons (interaction *p*-values > 0.20, Figs. 3–6, Supplementary Tables 16, 17). There were no intervention effects in any age or sex subgroup (Figs. 3–6, Supplementary Tables 16, 17). In both countries, effect modification estimates on multiplicative and additive scales yielded similar conclusions.

## Secondary and sensitivity analyses

Unadjusted and adjusted effect estimates were similar (Supplementary Tables 10–13). In sensitivity analyses, all interventions in Bangladesh reduced the percent of children who used antibiotics at least once in the last month by 10–16% (Supplementary Table 18). The nutrition and N + WSH interventions also appeared to reduce antibiotic use in the last two weeks by 7–11% but the associations could not be distinguished from chance, which could be due to reduced precision

**Table 1 | Caregiver-reported antibiotic use in last 3 months among young children enrolled in the control group of the WASH Benefits Bangladesh and Kenya trials**

|  | Bangladesh | | Kenya | |
|---|---|---|---|---|
|  | N | % (n) / mean (SD) | N | % (n) / mean (SD) |
| Used antibiotics ≥ 1 time | 1005 | 63.2 (635) | 1143 | 52.6 (601) |
| Used antibiotics ≥ 2 times | 1005 | 24.7 (248) | 1143 | 13.3 (152) |
| Episodes of antibiotic use | 1005 | 0.98 (0.98) | 1143 | 0.68 (0.75) |
| Total days of antibiotic use | 995 | 5.06 (5.81) | 1132 | 2.93 (3.97) |

*SD* Standard deviation

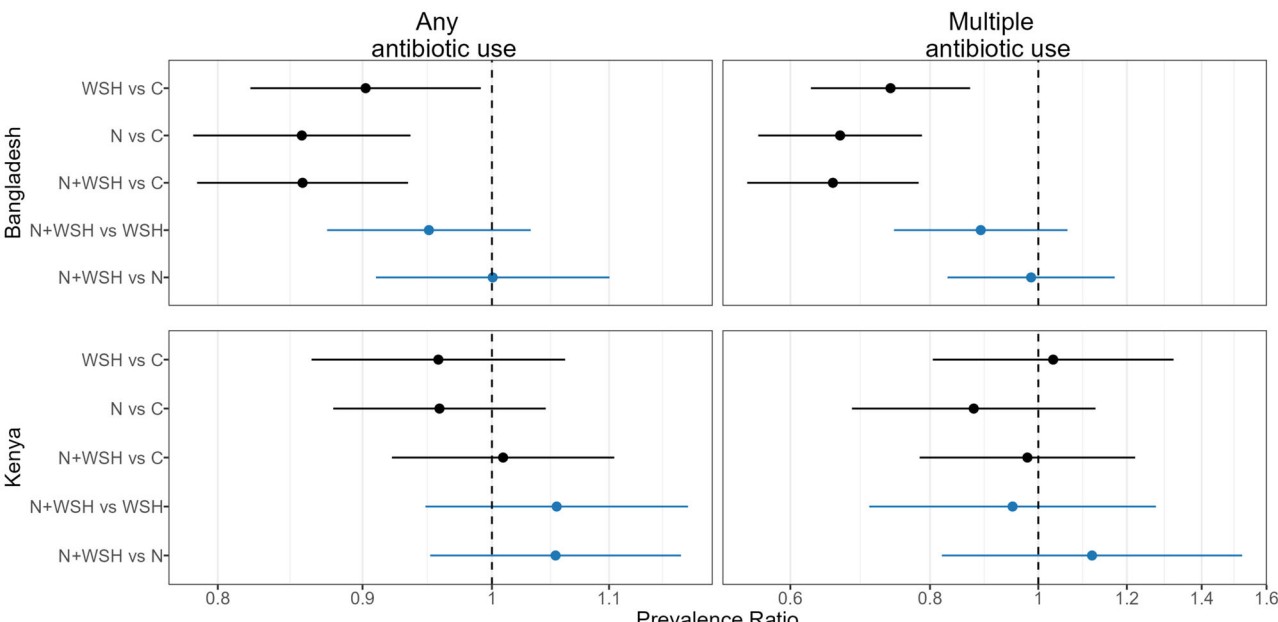

**Fig. 1 | Relative effects of water, sanitation, handwashing (WSH), nutrition (N) and nutrition plus WSH (N + WSH) interventions on the caregiver-reported prevalence of using antibiotics at least once and more than once in the last 3 months among young children in Bangladesh (N = 4158) and Kenya (N = 4280).** Estimates in black denote comparisons against the control (C) group who received no intervention. Estimates in blue denote comparisons of the N + WSH intervention group to the WSH and N intervention groups. Circles denote point estimates for prevalence ratios and horizontal lines denote 95% confidence intervals. Source data are provided as a Source Data file.

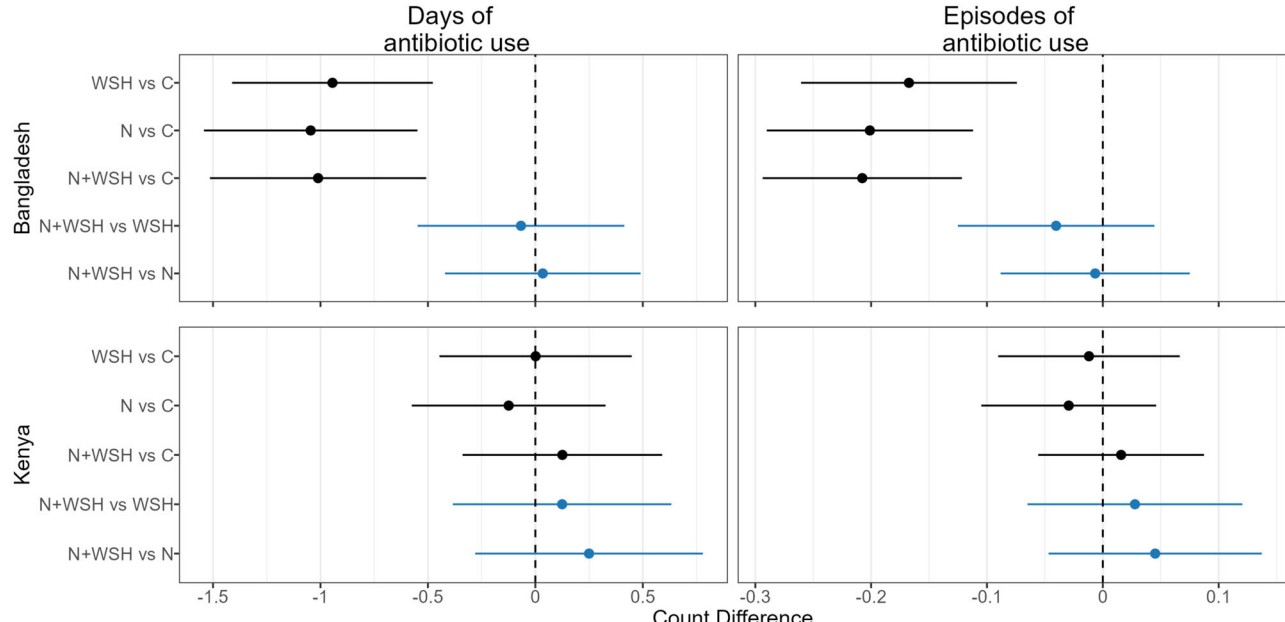

**Fig. 2 | Absolute effects of water, sanitation, handwashing (WSH), nutrition (N) and nutrition plus WSH (N + WSH) interventions on the caregiver-reported number of days and episodes of antibiotic use in the last 3 months among young children in Bangladesh (N = 4158) and Kenya (N = 4280).** Estimates in black denote comparisons against the control (C) group who received no intervention. Estimates in blue denote comparisons of the N + WSH intervention group to the WSH and N intervention groups. Circles denote point estimates for count differences and horizontal lines denote 95% confidence intervals. Source data are provided as a Source Data file.

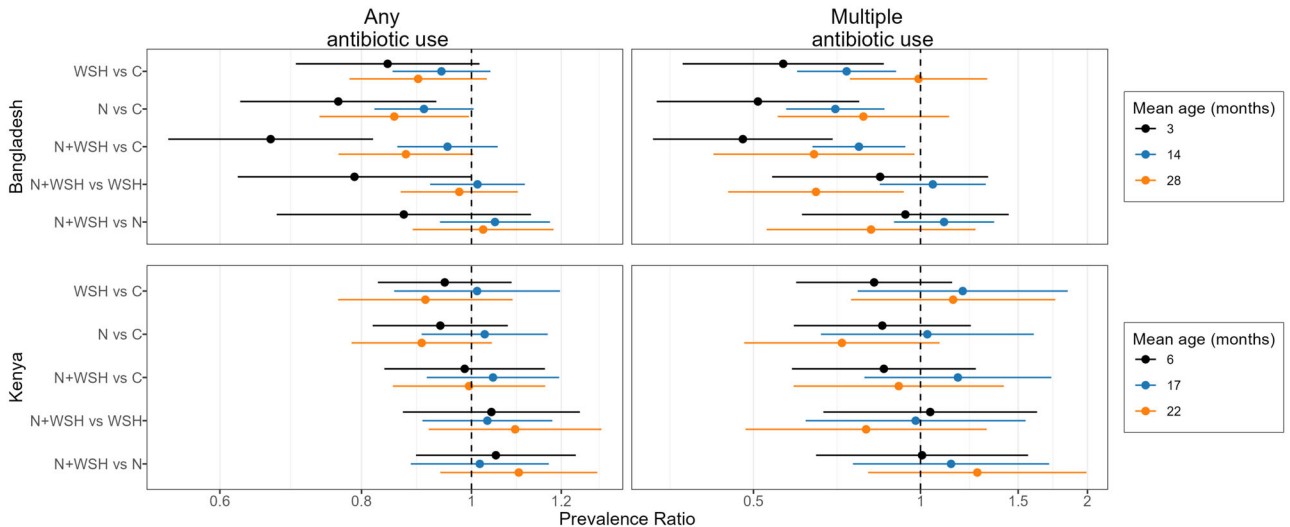

**Fig. 3 | Subgroup analysis by mean child age for relative effects of water, sanitation, handwashing (WSH), nutrition (N) and nutrition plus WSH (N + WSH) interventions on the caregiver-reported prevalence of using antibiotics at least once and more than once in the last 3 months among young children in Bangladesh and Kenya.** The control group (C) received no intervention. Antibiotic use was recorded when the children were on average 3 months (N = 1102), 14 months (N = 1528) and 28 months (N = 1528) old in Bangladesh, and 6 months (N = 1438), 17 months (N = 1449) and 22 months (N = 1393) old in Kenya. Circles denote point estimates for prevalence ratios and horizontal lines denote 95% confidence intervals. Source data are provided as a Source Data file.

because of the lower prevalence (~20%) of antibiotic use during this shorter window (Supplementary Table 18).

## Discussion

More than half of children enrolled in both Bangladesh and Kenya were reported to have used antibiotics at least once in the previous 90 days. Caregiver-reported antibiotic use was reduced among children randomized to receive WSH, nutrition and nutrition+WSH interventions in Bangladesh but not in Kenya. Combining WSH and nutrition interventions did not reduce caregiver-reported antibiotic use more than WSH or nutrition interventions alone. Our findings are broadly consistent with a recent double-blind, randomized controlled trial in urban Bangladesh that found that community-scale chlorination of drinking water reduced caregiver-reported antibiotic use by children in the past two months by 7%, along with a 23% reduction in diarrhea prevalence[20].

In Bangladesh, the interventions reduced caregiver-reported antibiotic use most at the first measurement when children were on average 3 months old. For this age group, the nutrition intervention included recommendations for maternal dietary diversity from pregnancy through lactation, early initiation of breastfeeding and exclusive

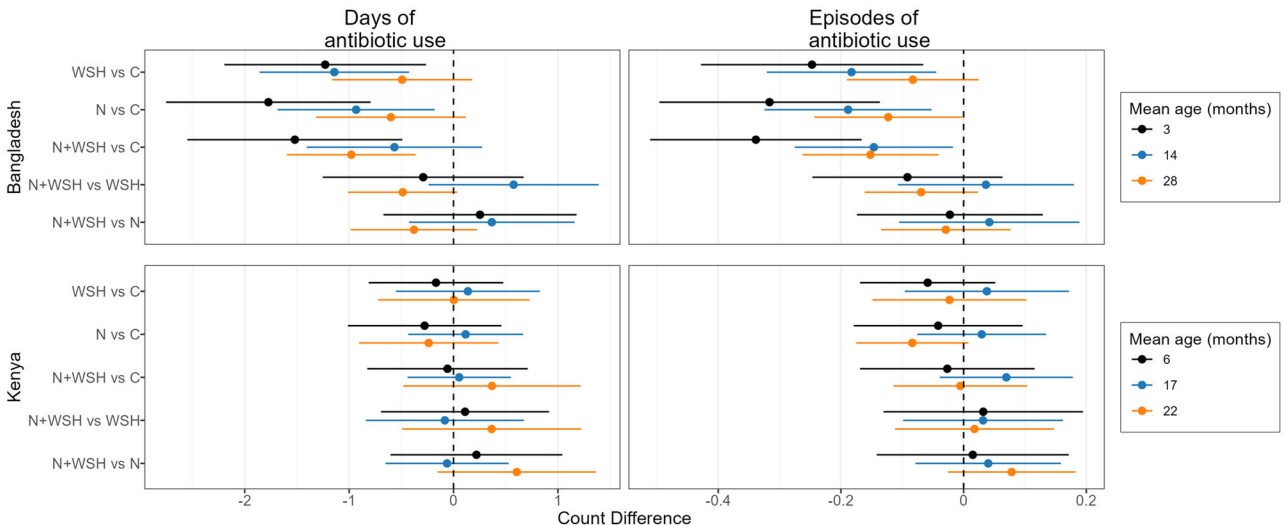

**Fig. 4 | Subgroup analysis by mean child age for absolute effects of water, sanitation, handwashing (WSH), nutrition (N) and nutrition plus WSH (N + WSH) interventions on the caregiver-reported number of days and episodes of antibiotic use in the last 3 months among young children in Bangladesh and Kenya.** The control group (C) received no intervention. Antibiotic use was recorded when the children were on average 3 months (N = 1102), 14 months (N = 1528) and 28 months (N = 1528) old in Bangladesh, and 6 months (N = 1438), 17 months (N = 1449) and 22 months (N = 1393) old in Kenya. Circles denote point estimates for count differences and horizontal lines denote 95% confidence intervals. Source data are provided as a Source Data file.

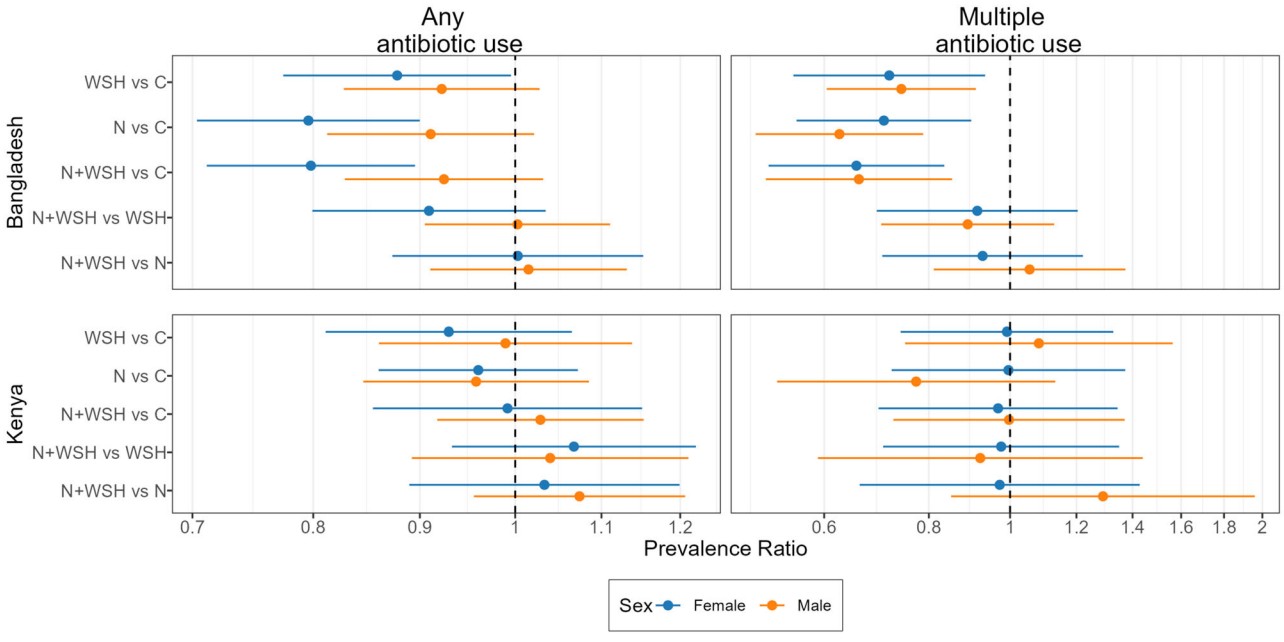

**Fig. 5 | Subgroup analysis by child sex for relative effects of water, sanitation, handwashing (WSH), nutrition (N) and nutrition plus WSH (N + WSH) interventions on the caregiver-reported prevalence of using antibiotics at least once and more than once in the last 3 months among young children in Bangladesh and Kenya.** The control group (C) received no intervention. Child sex was reported by the caregiver (Bangladesh N = 2082 girls, 2076 boys; Kenya N = 2208 girls, 2072 boys). Circles denote point estimates for prevalence ratios and horizontal lines denote 95% confidence intervals. Source data are provided as a Source Data file.

breastfeeding until 6 months, while effects of water, sanitation and hygiene interventions may be mediated through cleaner caregiver hands and a more hygienic domestic environment. Notably, in this age group, all three interventions were associated with a 28-44% reduction in caregiver-reported acute respiratory infections among children enrolled in the EED subset. In Bangladesh, interventions reduced the prevalence of using antibiotics at least once in the last 90 days among girls but not boys. This may reflect biological differences, sex-specific behaviors, or differential treatment by caregivers. For example, a recent meta-analysis found that newborn girls exhibited greater growth improvements from prenatal small-quantity lipid-based

nutrient supplements compared to boys[25]. In a birth cohort study in eight countries, girls were slightly less likely to receive antibiotics for diarrheal and respiratory infections than boys[7]. However, we observed similar intervention effects on the prevalence of using antibiotics multiple times, and the total times and days of antibiotic use for girls and boys, suggesting no overall trends by sex.

The observed reductions in antibiotic use demonstrate internal consistency with the trials' previously reported intervention effects on diarrhea and respiratory illness. In Bangladesh, diarrhea was reduced by 31-38% in the WSH, nutrition and N + WSH arms compared to controls[21], while respiratory infections were reduced by 33% in the

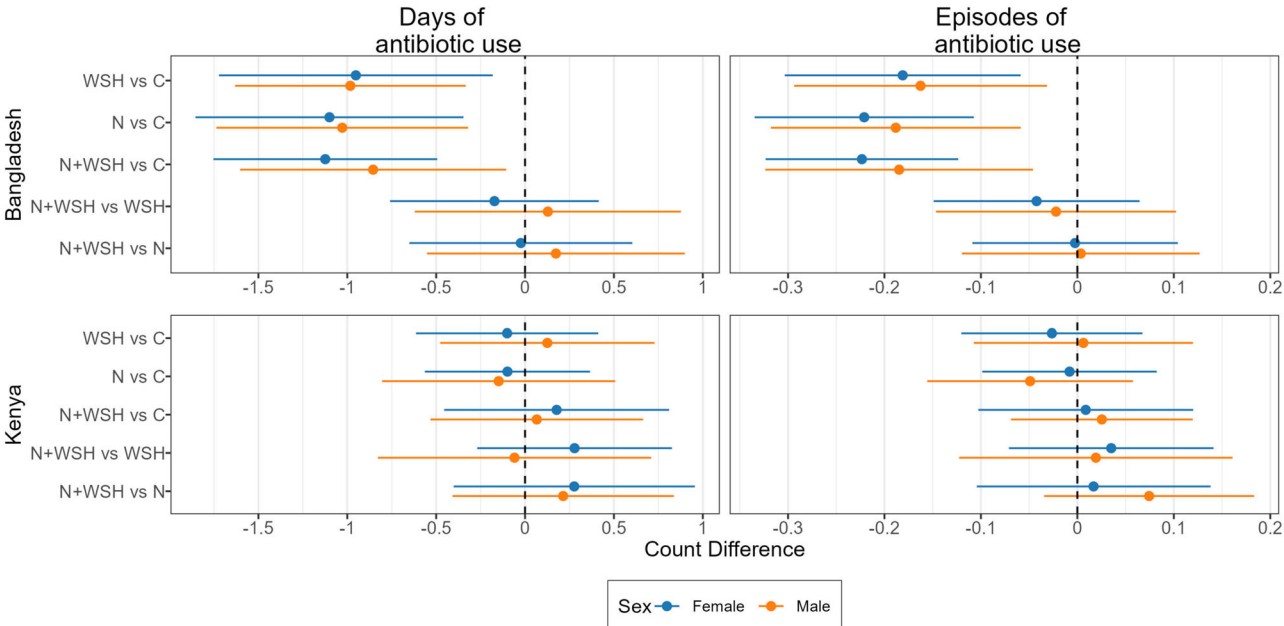

**Fig. 6 | Subgroup analysis by child sex for absolute effects of water, sanitation, handwashing (WSH), nutrition (N) and nutrition plus WSH (N + WSH) interventions on the caregiver-reported number of days and episodes of antibiotic use in the last 3 months among young children in Bangladesh and Kenya.** The control group (C) received no intervention. Child sex was reported by the caregiver (Bangladesh N = 2082 girls, 2076 boys; Kenya N = 2208 girls, 2072 boys). Circles denote point estimates for count differences and horizontal lines denote 95% confidence intervals. Source data are provided as a Source Data file.

N + WSH arm but not the nutrition and WSH arms[22]. Interventions also reduced diarrhea and respiratory infections among the subset of children enrolled in the EED substudy. Additionally, at the 14-month measurement point in the EED cohort in Bangladesh, TaqMan Array Card analysis of stool samples found that children in the WSH, nutrition and N + WSH groups carried fewer viruses, and children in the WSH group carried fewer total pathogens than controls but there was no effect on carriage of bacterial pathogens in any intervention group[26]. We found lower antibiotic use in all three of these study arms. Additional analysis demonstrated that the reductions in antibiotic use were mediated by reductions in the prevalence of reported diarrhea, reported acute respiratory infection with fever, and detection of enteric viruses in child stool[27]. Mediation through reduced respiratory infections (often of viral etiology) and carriage of enteric viruses suggests that interventions may have reduced uncalled-for antibiotic use prompted by viral infections, consistent with previous evidence that antibiotics are often unnecessarily prescribed for child diarrhea and respiratory infections in LMICs[7,8]. In Kenya, the interventions had no effect on diarrhea[23], and respiratory infections were reduced by 13% in the nutrition arm during the first year of the trial but not during the full study period and not in the WSH and N + WSH arms[24]. The interventions did not reduce diarrhea or respiratory infections among the subset of children enrolled in the EED substudy. Similarly, there were no intervention effects on antibiotic use.

One possible explanation for different intervention effects in Bangladesh vs. Kenya is differential adherence. High intervention uptake was sustained throughout the study in Bangladesh[28], whereas in Kenya uptake was lower and further decreased later in the trial[23]. In Bangladesh, in the WSH and N + WSH intervention groups, structured observations recorded defecation in hygienic latrines for 95–97% of adults, 85% of households had water and soap in latrine or kitchen areas, and 50–65% had detectable chlorine in their stored drinking water. In nutrition and N + WSH intervention groups, >80% of caregivers reported that children consumed the recommended amount of lipid-based nutrition supplements provided by the trial. In Kenya, 79–90% of WSH and N + WSH intervention households had improved latrines. Over 75% of households in these groups had water and soap at

the handwashing location in the first year and ~20% in the second year of the trial. Approximately 40% of WSH and N + WSH intervention households had chlorine in their water in the first year and 20% in the second year of the trial. Adherence to consumption of recommended lipid-based nutrition supplements was > 95% in the nutrition and N + WSH groups in both years of the trial. The observed reductions in antibiotic use in our analysis are consistent with these patterns in uptake and corroborate the importance of high intervention uptake in achieving effects on immediate as well as downstream outcomes.

The lack of intervention effects on antibiotic use in Kenya could also indicate that the household-level WSH and nutrition interventions did not address dominant drivers of infectious disease transmission in this setting. These could include contaminated food sources, domestic animals, and poor community-wide sanitation[29,30]. Alternatively, if antibiotic use is motivated by treating diarrhea and respiratory in infections in Bangladesh but by beliefs and preferences external to WSH and nutrition (e.g., to enhance child growth, treat other symptoms) in Kenya, differences in findings between the two countries may also be explained by differences in drivers of antibiotic use. Another possible explanation could be that the control arm in Bangladesh was passive (not visited by community health promoters) while the EED substudy in Kenya drew from the active control arm where community health promoters visited households regularly but did not promote any behaviors. Therefore, in Bangladesh, it is possible that additional interactions with the study team among intervention households influenced participant behaviors in addition to the interventions, potentially leading to stronger effects compared to passive controls. However, in Kenya, the primary trial outcomes were indistinguishable between the active and passive control arms[23], indicating no standalone effect from interaction with promoters.

Effects of WSH and nutrition interventions on community carriage of AMR need to be evaluated as an additional downstream outcome. Repeated antibiotic use within a short period persistently alters the gut microbiota into a predominantly resistant population by exerting selective pressure in favor of resistant strains[31]. Children in LMICs often carry enteropathogens asymptomatically in their gut; when antibiotics

are used to treat symptomatic respiratory and diarrheal infections, these "bystander" pathogens that are not the target of treatment are exposed to antibiotics, increasing the risk of resistance[32]. We would expect the observed reductions in antibiotic use among children receiving interventions in our analysis to translate to reduced carriage of AMR. WASH interventions can additionally directly interrupt the environmental spread and transmission of antimicrobial resistant organisms[33–35]. Nadimpalli et al.[36] found that community-scale water chlorination did not reduce antimicrobial resistance genes in child stool samples in urban Bangladesh despite reducing child diarrhea and reported antibiotic use, indicating that improving water quality alone in a setting with widespread contamination was not sufficient to reduce AMR. It has been suggested that poor sanitation[3] and environmental spread of antimicrobial resistant organisms[37] are stronger drivers of the global spread of AMR than antibiotic use. In a study of human gut metagenomes from 26 countries, access to improved drinking water and sanitation was associated with lower abundance of antimicrobial resistance genes[38]. Notably, antibiotic use for domestic animals and zoonotic transmission of antimicrobial resistant organisms are important contributors to community carriage of AMR in LMICs where humans and animals often share living spaces and animal fecal waste is not safely managed[39]. To achieve reductions in AMR, interventions targeting improved animal husbandry practices may be critical in addition to WASH improvements. Diet is also associated with the occurrence of antimicrobial resistance genes in the gut[40], and nutrition improvements have been proposed to reduce AMR[41].

Our findings support recommendations that future studies should assess the effect of WASH and nutrition interventions on community carriage of antimicrobial resistant organisms and antimicrobial resistance genes[42]. These assessments can focus on sentinel organisms such as extended-spectrum beta-lactamase producing *E. coli*, following WHO recommendations for global AMR surveillance[43]. Intervention studies should also collect data on antibiotic use as an outcome, which is cheaper and easier than measuring AMR in human biospecimens. There are no standardized data collection tools for recording antibiotic use. Studies can focus on predominantly used antibiotics in a given setting (e.g., based on pharmacy sales), and future work should explore best practices for developing and harmonizing data collection with respect to antibiotic classes of significance, optimal recall window for reporting use, and validation of self-reported use against medical records.

Additional potential downstream outcomes include effects on the microbiome and long-term sequelae. Diarrhea and antibiotic use are associated with alterations of gut microbiota and diminished microbiome richness and diversity[44–46]. Children exposed to antibiotics are at increased risk of a range of conditions, including asthma, juvenile arthritis, type 1 diabetes, Crohn's disease and mental illness; antibiotic-caused perturbations of the microbiome are believed to drive these risks[46,47]. Interventions that reduce early life infections and antibiotic use can support the natural maturational development of microbiome composition from infancy through childhood and potentially offer protection against these sequelae. Notably, we observed the largest reductions in antibiotic use from the interventions among the youngest children (mean age 3 months) in our study. The gut microbiome experiences rapid changes between the ages of 3–14 months, undergoes transition between 15-31 months and starts to stabilize into a mostly adult-like composition between 31–36 months[48]; minimizing perturbations to microbiota during these early life windows may deliver long-term health benefits.

A limitation of the study is that we relied on caregiver-reported antibiotic use, which is subject to poor recall and/or biased reporting given our non-blinded intervention. A previous study of birth cohorts in eight countries found good agreement between medical reports and caregiver-reported antibiotic use in children recorded via twice-weekly visits[7]. The observed concordance with medical records may not apply to the longer recall period in our study, and the possibility remains that the reported reductions in antibiotic use in Bangladesh were influenced by courtesy bias or placebo effects. However, our findings are internally consistent and biologically plausible; we observed reduced antibiotic use when high intervention uptake and reductions in diarrhea and respiratory infections were achieved. Our findings are also consistent with reductions in objectively measured outcomes in the Bangladesh trial. Children receiving WSH and N + WSH interventions had 17-25% lower carriage of Giardia[49] and 29-33% lower carriage of hookworm[50] in stool, compared to controls. Children receiving the WSH intervention had 49-65% lower carriage of enteric viruses (norovirus, sapovirus, adenovirus), and those receiving the nutrition intervention had 42% lower carriage of sapovirus[26]. Further, the reductions in virus carriage were found to mediate the observed reductions in antibiotic use[27]. Taken together, these findings lend support to a causal interpretation of intervention effects on caregiver-reported antibiotic use. Finally, we did not correct for multiple hypothesis testing because Bonferroni corrections and other multiplicity adjustments can lead to overcorrections[51]. Therefore, some of the reported effects could be chance findings, but observed effects were highly consistent across different analyses and unlikely to be explained by chance.

A strength of the study is evidence from two high-risk settings and cluster-randomized allocation of interventions with over 150 clusters in each trial's substudy. The trials were efficacy trials with free provision of products and intensive promotion so the same effects may not be achieved when interventions are programmatically implemented at scale – nevertheless, the efficacy trials provide evidence that an effect on antibiotic use is possible with intensive WASH and nutritional interventions that reduce clinical illness among young children. We also note that the delivery mechanism for nutrition supplements was through home visits by project staff. Mass distribution of such supplements by healthcare systems can have different effects on antibiotic use. For example, adding nutrition supplements into health programs can increase program participation[52,53]. While this can lead to higher immunization rates, which would reduce antibiotic use by reducing illness[54], increased contact with the health system can also increase unnecessary antibiotic prescriptions.

Global efforts to limit AMR primarily focus on limiting inappropriate antibiotic use, providing appropriate antimicrobials as needed, improving WASH access, and reducing exposure to untreated waste[55,56]. While antibiotic stewardship can limit the unnecessary use of antibiotics and subsequent AMR[57,58], our results provide support for upstream WASH and nutrition interventions to reduce antibiotic use through decreased infection. Studies should assess the effects of WASH and nutrition interventions on community carriage of AMR.

## Methods

### Inclusion and ethics

Primary caregivers of children provided written informed consent. The study protocol was approved by human subjects committees at the International Centre for Diarrhoeal Disease Research, Bangladesh (PR11063), Kenya Medical Research Institute (SSC-2271), University of California, Berkeley (2011-09-3652), and Stanford University (25863).

### Study design and participants

The WASH Benefits trials were cluster-randomized controlled trials that enrolled pregnant women in rural Bangladesh and Kenya. The Bangladesh trial (NCT01590095) was conducted in contiguous rural subdistricts in Gazipur, Mymensingh, Tangail and Kishoreganj districts of central Bangladesh. The Kenya trial (NCT01704105) was conducted in rural villages in Kakamega, Bungoma, and Vihiga counties in western Kenya. The study areas were chosen to have no other ongoing WASH/nutrition programs. Details of the study design have been described[59].

## Randomization and masking

Field staff screened the areas to enroll pregnant women over the course of approximately one year. Geographically matched clusters of 6–8 compounds where the enrolled pregnant women lived were block-randomized to control or one of the intervention arms by an off-site investigator using a random number generator. The cluster design was chosen to minimize between-arm spillovers and facilitate intervention delivery logistics. Participants or data collectors were not blinded because interventions included visible hardware.

## Procedures

Interventions were initiated in rolling fashion around when the enrolled women gave birth and included water treatment, sanitation, handwashing, nutrition, combined water treatment, sanitation and handwashing (WSH), and nutrition plus combined WSH (N + WSH). The water treatment component included chlorine tablets and a safe storage vessel provided to households in Bangladesh and chlorine dispensers installed at all community water locations plus bottled chlorine provided to households in Kenya. The sanitation component included double-pit latrines for all households in study compounds in Bangladesh, pit latrine upgrades with a reinforced slab and drop hole cover in Kenya, and child potties and hoes for feces management in both countries. The handwashing component included handwashing stations with soapy water solution and rinse water near the latrine and kitchen. The nutrition component included lipid-based nutrient supplements (LNS) for the birth cohort and age-appropriate recommendations on maternal nutrition and infant feeding practices.

Intervention hardware and consumables were provided free of charge and replenished throughout the study period. Local promoters hired and trained by study staff visited study compounds regularly to promote: (1) treating drinking water for children aged <36 months, (2) use of latrines/child potties for defecation and removal of human and animal feces from the compound, (3) handwashing with soap at critical times around food preparation, defecation, and contact with feces, and (4) LNS for children aged 6–24 months and age-appropriate nutrition practices from pregnancy to 24 months. The promoters did not provide any medical treatment or advice. In Bangladesh, promoters visited participants six times per month on average. In Kenya, they were instructed to visit participants several times in the first two months while interventions were delivered, monthly for the rest of the first year, and every two months the second year. In Bangladesh, promoters did not visit control households (passive controls). In Kenya, the trial included both passive and active control arms to isolate the influence of engagement with the study team from intervention effects. Promoters did not visit passive control households. They visited active control households monthly to measure child mid-upper arm circumference but did not promote any behavior change. User uptake of targeted behaviors was assessed by structured and spot check observations. Uptake was high and sustained in Bangladesh[28,60,61] but lower and variable in Kenya[23]. Details of intervention delivery and uptake have been described[23,28,60,61].

Caregiver-reported antibiotic use was recorded among children participating in a substudy conducted to assess environmental enteric dysfunction (EED). The EED substudy was conducted in nutrition, WSH, N + WSH and control arms (passive in Bangladesh, active in Kenya) of the parent trial. To facilitate collection of biological specimens, enrollment in the substudy focused on areas close to the field laboratory and did not follow the geographic matching of the parent trial. The EED substudy in Kenya was limited to the Kakamega and Bungoma counties and excluded children <1 month old without a clinic card and children <2 weeks old due to lack of parental consent. The EED substudy enrolled 267 clusters in Bangladesh and 190 clusters in Kenya and aimed to enroll 1500 children (375 per arm) from the birth cohort per country.

For children in the EED substudy, we recorded caregiver-reported antibiotic use in three longitudinal rounds in approximately one-year intervals. Due to challenging logistics and political unrest, we were not able to synchronize child ages at follow-up across the countries; children were on average 3, 14, and 28 months old in Bangladesh and 6, 17, and 22 months old in Kenya at the three measurement points. Trained field staff asked the primary caregiver how many times the child used antibiotics within 90 days before the visit, and the number of days the child used specific antibiotics during this window. When available, field staff asked for the prescription or the packaging for the antibiotic, and otherwise asked caregivers to recall the name of the antibiotic from among a list of 11 antibiotics commonly used in the study regions (cotrimoxazole, amoxicillin, flucloxacillin, ciprofloxacin, erythromycin, azithromycin, nalidixic acid, doxycycline, penicillin, chloramphenicol, metronidazole). If the taken antibiotic was not listed, field staff asked caregivers to specify the antibiotic or else marked it as unknown. We did not record whether the caregiver was able to produce the prescription or packaging for the antibiotic. We also did not record whether the antibiotic was taken therapeutically or prophylactically, whether it was prescribed and where it was obtained.

## Outcomes

For the present analysis, our pre-specified primary outcome was the prevalence of children who used antibiotics at least once within 90 days prior to data collection, tabulated individually at each of the three measurements points. Pre-specified secondary outcomes were the prevalence of children who used antibiotics multiple times, number of episodes of antibiotic use, and number of days of antibiotic use, within 90 days prior to data collection. We compared these outcomes in each intervention arm against controls and in the N + WSH arm against the nutrition and WSH arms. We also compared the 7-day prevalence of diarrhea and acute respiratory infections in intervention vs. control arms to assess whether the intervention effects on these outcomes reported by the parent trials were observed among the subset of children in the EED substudy.

## Statistical analysis

**Hypotheses.** We hypothesized that (1) children receiving nutrition, WSH or N + WSH interventions would have reduced antibiotic use compared to controls and (2) children receiving N + WSH interventions would have reduced antibiotic use compared to those receiving WSH or nutrition interventions alone. These hypotheses follow the pre-specified hypotheses of the WASH Benefits trials for the primary trial outcomes.

**Estimation strategy.** We conducted comparisons separately for each country, using pooled data from all three measurement points. Analyses were intention-to-treat. We estimated prevalence ratios (PRs) and prevalence differences (PDs) for binary outcomes, and count ratios (CRs) and count differences (CDs) for count outcomes. We used generalized linear models with robust standard errors to account for geographical clustering and repeated measurements. We used a Poisson error distribution and log link to estimate PRs for binary outcomes[62], Poisson or negative binomial error distribution and log link to estimate CRs for count outcomes, and a Gaussian error distribution and identity link to estimate PDs and CDs. Randomization led to good balance in measured covariates across arms[21,23] so our primary inference focused on unadjusted comparisons between groups. In additional analyses, we estimated adjusted effects by including variables strongly associated with the outcome to potentially improve the precision of our estimates[63]. As per the analysis plan of the WASH Benefits trial[59] (updated on 2016.02.05, https://osf.io/63mna/), we considered the following adjustment covariates, recorded either at the time of outcome ascertainment (date in 3-month intervals, child age and sex) or at the trial's baseline (birth order, mother's age, height and education, household food insecurity, number of individuals <18 years in household, number of individuals living in compound, distance to household's primary drinking water source, housing materials, and

household wealth index calculated from principal components analysis of household assets). We used likelihood ratio tests to assess the association between each covariate and outcome and included covariates with a p-value <0.20 in adjusted analyses. Details of our analysis approach are available in a pre-specified analysis plan, along with de-identified datasets and analysis scripts (https://osf.io/t7fmw/). Analyses were conducted using R version 4.0.3 GUI 1.73 and the R package "washb" that was developed to standardize analyses of the WASH Benefits trial data. Information on the package is available (https://ben-arnold.github.io/washb/articles/washb.html).

**Statistical power.** We calculated the minimum detectable effects (MDEs) based on the prevalence of control children who used antibiotics at least once in the 90 days (63% in Bangladesh, 53% in Kenya), number of observations per study cluster (17 in Bangladesh, 26 in Kenya), and the intracluster correlation coefficient for observations in the same cluster (0.04 in Bangladesh, 0.03 in Kenya). Our sample size yields 80% power with a two-sided alpha of 0.05 to detect the following MDEs between any intervention arm vs. controls: 11% relative reduction in Bangladesh and 13% relative reduction in Kenya in the prevalence of using antibiotics at least once in 90 days.

**Effect modification.** Children of different age groups have different physiological characteristics, levels of immunity and risk of infection[64]. In addition, different stages of breastfeeding, weaning, mobility and dexterity result in varying exposure to pathogens through contaminated water, food, hands and objects[65–67]. Child sex may also influence intervention delivery, immune status and antibiotic use[68–70]. We hypothesized that the effects of WSH and nutrition interventions on antibiotic use could vary with child age and sex. Within each country, we investigated effect modification by the three measurement points (corresponding to a mean child age of 3, 14, 28 months in Bangladesh, and 6, 17, 22 months in Kenya) and by sex. Child sex was reported by the caregiver; we did not differentiate between sex and gender given the young age groups. We included interaction terms between study arm and these effect modifiers in regression models and investigated both multiplicative and additive interaction. We interpreted interaction p-values <0.20 as evidence of effect modification.

**Sensitivity analyses.** It is possible that caregiver-reported antibiotic use within the last three months is subject to inaccurate recall, and a shorter recall window may increase the accuracy of reporting. As a sensitivity analysis, we estimated intervention effects on antibiotic use during the last month and last two weeks. These variables were derived from a separate question on how long ago the child last used antibiotics. In Kenya, this question recorded both antibiotics and non-antibiotic medications so the sensitivity analysis was only conducted for Bangladesh.

### Reporting summary
Further information on research design is available in the Nature Portfolio Reporting Summary linked to this article.

## Data availability
All data used in the presented analyses are publicly available at: https://osf.io/t7fmw/ Source data are provided with this paper.

## Code availability
All code used in the presented analyses is publicly available at: https://osf.io/t7fmw/.

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

## Acknowledgements

This research was financially supported by a global development grant (OPPGD759 to JMC) from the Bill & Melinda Gates Foundation to the University of California, Berkeley, CA, USA.

## Author contributions

A.E.: Conceptualization, data curation, formal analysis, methodology, writing—original draft. A.N.M.: Data curation, formal analysis, writing—review & editing. Z.B.D.: Visualization, writing—review & editing. D.K.J.: Investigation, writing—review & editing. S.A.: Supervision, data curation, investigation, writing—review & editing. BSA: Supervision, data curation, investigation, writing—review & editing. G.R.: Supervision, data curation, investigation, writing—review & editing. C.H.: Investigation, writing —review & editing. A.J.P.: Investigation, supervision, writing—review & editing. C.P.S.: Investigation, supervision, writing—review & editing. S.T.T.: Data curation, investigation, writing—review & editing. J.A.G.: Data curation, investigation, writing—review & editing. J.B.C.: Investigation, project administration, writing—review & editing. M.W.: Data curation, investigation, writing—review & editing. G.G.H.: Investigation, writing—review & editing. M.Z.R.: Data curation, investigation, writing—review & editing. C.D.A.: Data curation, investigation, writing—review & editing. H.N.D.: Supervision, investigation, writing—review & editing. S.M.N.: Investigation, supervision, writing—review & editing. T.M.: Supervision, investigation, writing—review & editing. B.C.: Data curation, investigation, writing—review & editing. M.N.: Writing—review & editing. M.A.I.: Writing—review & editing. A.E.H.: Methodology, writing—review & editing. C.N.: Investigation, supervision, writing—review & editing. L.U.: Investigation, supervision, writing—review & editing. M.R.: Investigation, supervision, writing—review & editing. J.M.C.: Conceptualization, funding acquisition, investigation, supervision, writing—review & editing. S.P.L.: Conceptualization, funding acquisition, investigation, supervision, writing—review & editing. B.F.A.: Conceptualization, funding acquisition, investigation, methodology, writing—review & editing. A.L.: Conceptualization, supervision, data curation, investigation, writing —review & editing.

## Competing interests

The authors declare no competing interests.

## Additional information

¹Department of Forestry and Environmental Resources, North Carolina State University, Raleigh, NC, USA. ²Division of Epidemiology and Biostatistics, School of Public Health, University of California, Berkeley, Berkeley, CA, USA. ³Environmental Health and WASH, Health System and Population Studies Division, International Centre for Diarrhoeal Disease Research, Bangladesh, Dhaka, Bangladesh. ⁴Innovations for Poverty Action, Nairobi, Kenya. ⁵Department of Environmental Sciences and Engineering, Gillings School of Global Public Health, University of North Carolina at Chapel Hill, Chapel Hill, NC, USA. ⁶Department of Civil and Environmental Engineering, Blum Center for Developing Economies, University of California, Berkeley, Berkeley, CA, USA. ⁷Chan Zuckerberg Biohub, San Francisco, CA, USA. ⁸Institute for Global Nutrition, University of California, Davis, Davis, CA, USA. ⁹Division of Infectious Diseases and Geographic Medicine, Department of Medicine, School of Medicine, Stanford University, Stanford, CA, USA. ¹⁰Department of Epidemiology and Population Health, School of Medicine, Stanford University, Stanford, CA, USA. ¹¹Rollins School of Public Health, Emory University, Atlanta, GA, USA. ¹²Kenya Medical Research Institute, Nairobi, Kenya. ¹³Paul G. Allen School for Global Health, Washington State University, Pullman, WA, USA. ¹⁴Mathematica Policy Research, Washington, DC, USA. ¹⁵Global Health and Migration Unit, Department of Women's and Children's Health, Uppsala University, Uppsala, Sweden. ¹⁶Francis I. Proctor Foundation and Department of Ophthalmology, University of California, San Francisco, San Francisco, CA, USA. ¹⁷University of California, Santa Cruz, Santa Cruz, CA, USA. ✉e-mail: aercume@ncsu.edu

