## [Transparent Peer Review file · Nature Communications]

Water, sanitation, handwashing, and nutritional interventions can reduce child antibiotic use: Evidence from Bangladesh and Kenya

Corresponding Author: Dr Ayse Ercumen

Version 0:

Reviewer comments:

Reviewer #1

(Remarks to the Author)

This manuscript reports on a novel analysis demonstrating how WASH and nutrition interventions prevented antibiotic use in a large two site clinical trial, presumably by preventing infections and improving child growth. It is a well-conducted and reported study that includes appropriate subgroup analyses and sensitivity analyses to account for recall bias in the outcome. Most comments are minor below. One limitation is that while the authors describe how their results are consistent with the intervention effects on other outcomes (diarrhea, resp infections, enteric infections), they do not directly assess whether the effects on antibiotic use are mediated by reductions in these intermediate outcomes. Given this data is available in their trial dataset, this work would be considerably stronger if a formal mediation analysis was added to determine which reductions in relevant clinical outcomes (or improvements in growth) at the individual level are responsible for the intervention effects on antibiotic use.

Specific comments include:

1. Abstract line 47 – specify the time period here (in the last 90 days of what? specify 3 cross-sectional measurements).
2. In the background it would be helpful to report the intervention effects in the EED substudy to confirm whether they were the same or different from the original trial; if different, the implications for this analysis should be discussed
3. Procedures like 343 – this statement is not clear: “Interventions were initiated around when the birth cohort was born.” What birth cohort? For the whole study? For each cluster? Babies were all born at the same time?
4. Procedures line 383/384 – in addition to the average ages, it would be helpful to provide the variability in measurement ages (how long were these cross-sectional surveys?)
5. Procedures outcomes – is the outcome at any of the 3 time points? Or any use over any of the 3 time windows? (specified later in the analysis section, but should be included here)
6. Procedures line 422 – “data collection date” is the outcome collection date or something else? Child age is age at outcome data collection? Everything else at baseline? Or at time of outcome ascertainment?
7. Results lines 129/137 – this is minor, but should just report the prevalence in boys and girls in the text here
8. Results line 153 – report the p value for the heterogeneity test here
9. Results line 196 – this interpretation is a little misleading. The effects are very similar in last month/last 2 weeks, but last 2 weeks is not significant, probably because smaller number with outcome (less precision), not because effect is not there
10. Because some outcomes are on the absolute and some on the relative scale, additive and multiplicative interaction are both assessed which is great. This is noted in the methods but glossed over in the results – does the evidence for multiplicative and additive interaction always agree? If not, where do they disagree?
11. Discussion line 215- can you do a mediation analysis to assess whether the reductions observed in diarrhea/resp illnesses (or viral infections, or intervention adherence) are responsible for reductions in antibiotic use?
12. Discussion line 297 – was there agreement between caregiver report and medical report even with a long recall period like 90 days?
13. Discussion – can the authors comment more on why there were observed differences by sex?

(Remarks on code availability)

Code and data are not currently publicly available on the provided site. Therefore I was unable to run the code.

Reviewer #2

(Remarks to the Author)

This is a secondary data analysis of WASH Benefits trial data from Bangladesh and Kenya, which mirrors the primary results of the trial – in a setting where there were reductions in diarrhoea (Bangladesh) there were modest reductions in antibiotic use; in a setting where there was no reduction in diarrhoea (Kenya), there was no reduction in antibiotic use.

Antibiotic data were available for a subgroup of children participating in an EED substudy, who had similar characteristics to the broader trial population. My major comment is that there are a huge number of analyses presented, including subgroup analyses and sensitivity analyses, but these are all underpowered as secondary/tertiary exploratory analyses, and with no adjustment for multiple hypothesis testing. In particular, the sensitivity analyses show that the inferences change if the recall period for antibiotics is 2 weeks, rather than 1 month of 90 days, but one would expect that recall of antibiotic use is actually most reliable in the prior 2 weeks, and much less so in the 30-90 days prior.

A few other comments:

1. Were antibiotics used for prophylaxis excluded? For example, in Kenya where HIV-exposed children were likely prescribed cotrimoxazole throughout the period of breastfeeding?
2. Where were antibiotics obtained from? Did the trial provide treatment of illness?
3. Contacts between the study team and the households were not the same in the intervention and control arms. Did fieldworkers provide advice about management of illness during their visits to active arms, and thereby influence behaviours of caregivers potentially? In which case, this reduction would be more due to behaviour change communication, rather than WASH per se.
4. The major effects seen in Bangladesh were at 3 months of age – what did the nutrition intervention comprise at this age, and what about WASH at this young age?
5. Some statements need revising to be more nuanced: eg “immune systems weakened by poor nutritional status put children at risk of further infections”; and “Repeated antibiotic use within a short period persistently alters the gut microbiota into a predominantly resistant population”.

(Remarks on code availability)

Reviewer #3

(Remarks to the Author)

***What are the noteworthy results?

This study reanalyzes data from the WASH Benefits trial that measured the impact of improvement in water, sanitation, and hygiene (WSH), nutrition (N), or both (WSH+N) on a variety of environmental, behavioral, and clinical health outcomes. In this case, the study examines the impact of the trial on caregiver-reported antibiotic usage in children in both Bangladesh and Kenya sites overall and by age group. Noteworthy results include demonstration of high levels of reported antibiotic use for multiple types of antibiotic classes between birth and 28 months of age and demonstration of similar country-level patterns of impact of WSH, N, and WSH+N interventions on reported antibiotic use as observed in analysis of trial impact on self-reported diarrhea and respiratory symptoms. This study also finds interaction between trial reductions in usage by age with strongest impact on the youngest ages.

***Will the work be of significance to the field and related fields? How does it compare to the established literature? If the work is not original, please provide relevant references.

While studies on antibiotic use are common, evidence on how WSH and nutrition interventions impact antibiotic use are relatively few in number and WASH Benefits is considered a powerful study for examining causal relationships between WSH and nutrition conditions and early childhood health. This study will be well received by the science community.

***Does the work support the conclusions and claims, or is additional evidence needed? Are there any flaws in the data analysis, interpretation and conclusions? Do these prohibit publication or require revision?

Part of the work supports the conclusions and claims although the discussion lacks some critical points about study design factors that could explain the inconsistent study results between Bangladesh and Kenya and thus the need for caution in inference about generalizability of WSH+N impacts on antibiotic usage across different settings. Recommended edits prior to publication are as follows:

MAJOR COMMENTS

1. Authors address the limitation of using self/caregiver-reported antibiotic usage in the discussion. However, the methods

suggest that they may have an observable outcome that could be compared to reported outcomes as an assessment of potential reporting bias. Specifically, the methods state that if participants reported antibiotic usage, then field staff asked to see the prescription or packaging for the prescription. At minimum, the proportion of caregivers in each group in each site who produced an antibiotic or package for inspection should be reported in Table 1 and in the Results paragraph on usage. If sufficient data is available, it would also be valuable as an outcome in models. This would allow for a more nuanced discussion on reliability of the reported antibiotic usage outcome in the discussion, and potentially even bolster the self-reported outcome results.

2. Page 10 includes a thorough proposed explanation for different effects of the WSH, N, and WSH+N intervention in Bangladesh versus Kenya. Another possible explanation that needs to be acknowledged and discussed after this paragraph is the potential for differential courtesy bias caused by study design. The passive control approach in Bangladesh vs active control approach in Kenya meant controls had different levels of contact with the study team. Kenyans with greater contact with the study team may have over or under reported relative to Bangladeshi participants based upon knowledge and reaction to being closely studied - such as altering responses based upon beliefs about what responses might be expected or desired of them. Similarly, Bangladeshi trial participants with promoters visiting six times per month may have over or under-reported usage based upon knowledge of trial status and perception that certain types of responses may be expected of them relative to trial participants in Kenya. Alternatively, the observed differences in usage as well as self-reported diarrhea in Bangladesh may be factual but be more due to the impact of health promoters visiting participants so frequently and less about WSH and N intervention components. The fact that WASH-B impacted self-reported diarrhea and respiratory symptoms (and antibiotic usage) to a much greater extent than anything observed with objectively measured pathogen infections or environmental indicators in Bangladesh is concerning, given that the impact of the trial on reported and objective measures in Kenya was pretty consistent. Thus, study design should be acknowledged as a potential influence on trial effects, at least in the context of this study on reported antibiotic usage.

3. If your hypothesis about WSH+N impact on antibiotic usage being an outcome of WSH+N reductions in diarrhea/respiratory symptoms is correct, then in the discussion about why Kenya did not experience the same WSH+N impact on antibiotic usage, you could acknowledge that no reduction in usage would be expected if primary sources of infections in Kenya are from sources that are poorly addressed by the WSH+N design, for example food sources, flies, or animals.

4. In the discussion of limitations, please add that this study did not document the motivations for antibiotic usage so if usage is more motivated by beliefs and factors external to WSH or nutrition (e.g. use for other symptoms, beliefs that antibiotics enhance breastmilk quality or growth and development of the infant) then there should be no effect of WSH or N interventions on antibiotic use, even if WSH and N reduce diarrhea or respiratory symptom prevalence. The relationship between drivers and usage could also explain the differences observed between countries in this study if usage in Bangladesh is motivated by diarrhea or respiratory symptoms, by WSH and/or N knowledge, or by courtesy bias but in Kenya usage is motivated more by the factors cited above.

Minor comments

1. Several places in the manuscript read as if antibiotic usage was observed or validated, even if that was unintended. More consistent use of "reported antibiotic usage" or "caregiver-reported antibiotic usage" at key points such as in tables and figures, in the Introduction final paragraph, and at the beginning of the discussion is needed to ensure readers understand the nature of the outcome measure.

2. The last sentence of the first paragraph on page 3 is not correct because it has flipped the results of these two papers. References 6 and 7 measured antibiotic use in people with diarrhea and respiratory symptoms, not the primary reasons for antibiotic use. Reasons for antibiotic use are also commonly unrelated to diarrhea and respiratory symptoms, such as prescriptions for general fevers, rashes, or other symptoms, as well as self-medication choices based upon beliefs that antibiotics enhance physical growth, ensure healthy pregnancies and the quality of breastmilk, among other cultural beliefs.

3. On page 2 and 11, language references primary drivers of antimicrobial resistant emergence and transmission but does not acknowledge the importance of zoonotic sources. Many consider antibiotic use in animals a more important driver of AMR transmission and carriage than human sources so acknowledging zoonotic vectors in these two locations would be more holistic.

4. The end-of-paragraph sentence at the beginning of page 10 about viral vs bacterial differences in WASH-B trial impact should be followed by the subsequent observation that if trial arms did reduce antibiotic usage because of reduction in symptomatic viral infections, that lower antibiotic usage did not reduce or increase the probability of trial or control participants experiencing a bacterial infection.

5. Page 13, please add the usage recall period used in reference 6 to the statement "However, a previous study of birth cohorts in eight countries found good agreement between caregiver-reported antibiotic use in children and medical reports 6."

***Is the methodology sound? Does the work meet the expected standards in your field?

The approach to measurement of antibiotic use is consistent with standards in the field. Authors address the challenges in accurate measurement of usage and propose rigorous ways to improve that measurement in the future. The design of the WASH-B parent trial that was leveraged for this sub-study is rigorous. The analytical approach is standard but suited to the research hypothesis.

***Is there enough detail provided in the methods for the work to be reproduced?

The methods provide enough detail for the work to be reproduced, especially in combination with prior publications from the same authors on study design details.

(Remarks on code availability)

I have skimmed the code and it appears usable although i have not replicated the results due to lack of time to do so with rigor.

Version 1:

Reviewer comments:

Reviewer #1

(Remarks to the Author)

The authors have sufficiently responded to my comments.

(Remarks on code availability)

Reviewer #2

(Remarks to the Author)

I am satisfied with the author responses.

(Remarks on code availability)

Reviewer #3

(Remarks to the Author)

The authors were responsive to the critiques of all reviewers and the edits and added statements provided through that response, the added mediation analysis and acknowledgments of study design limitations in particular, have improved the transparency and rigor of reported study results. I have no further comments and believe the manuscript is suitable for publication.

(Remarks on code availability)

The read me files with instructions and code are accessible and the models encoded usable code.

Reviewer #1 (Remarks to the Author):

This manuscript reports on a novel analysis demonstrating how WASH and nutrition interventions prevented antibiotic use in a large two site clinical trial, presumably by preventing infections and improving child growth. It is a well-conducted and reported study that includes appropriate subgroup analyses and sensitivity analyses to account for recall bias in the outcome. Most comments are minor below. One limitation is that while the authors describe how their results are consistent with the intervention effects on other outcomes (diarrhea, resp infections, enteric infections), they do not directly assess whether the effects on antibiotic use are mediated by reductions in these intermediate outcomes. Given this data is available in their trial dataset, this work would be considerably stronger if a formal mediation analysis was added to determine which reductions in relevant clinical outcomes (or improvements in growth) at the individual level are responsible for the intervention effects on antibiotic use.

Response: Thank you for raising this helpful point about investigating mediation. We now report mediation effects. Please see our detailed response under comment #11 below.

Specific comments include:

1. Abstract line 47 – specify the time period here (in the last 90 days of what? specify 3 cross-sectional measurements).

Response: We have specified that we are referring to the 90 day-period before the data collection visit and that the data were collected through three longitudinal visits.

“We assessed effects of water, sanitation, handwashing (WSH) and nutrition interventions on antibiotic use in Bangladesh and Kenya, **longitudinally measured at three timepoints** among birth cohorts (ages 3-28 months) in a cluster-randomized trial. Over 50% of children used antibiotics at least once in the 90 days **preceding data collection**.”

2. In the background it would be helpful to report the intervention effects in the EED substudy to confirm whether they were the same or different from the original trial; if different, the implications for this analysis should be discussed

Response: This is a very helpful suggestion. We have added this additional analysis to our methods, results and discussion sections. The observed intervention effects among the subset of children enrolled in the EED subset were similar to effects observed in the parent trial.

“We also compared the 7-day prevalence of diarrhea and acute respiratory infections in intervention vs. control arms to assess whether the intervention effects on these outcomes reported by the parent trials were observed among the subset of children in the EED substudy.”

“In Bangladesh, among the subset of children enrolled in the EED study, those who received WSH and N+WSH interventions had 21-35% lower prevalence of diarrhea and those who

received N+WSH interventions had 21-44% lower prevalence of acute respiratory infections, compared to controls (Supplementary Table 6). Notably, at age 3 months, all three interventions reduced the prevalence of acute respiratory infections by 28-44% (Supplementary Table 6). In Kenya, there were no intervention effects on diarrhea and acute respiratory infections among the subset of children enrolled in the EED substudy (Supplementary Table 7).

“In Bangladesh, diarrhea was reduced by 31-38% in the WSH, nutrition and N+WSH arms compared to controls ²¹, while respiratory infections were reduced by 33% in the N+WSH arm but not the nutrition and WSH arms ²². **Interventions also reduced diarrhea and respiratory infections among the subset of children enrolled in the EED substudy.**”

3. Procedures like 343 – this statement is not clear: “Interventions were initiated around when the birth cohort was born.” What birth cohort? For the whole study? For each cluster? Babies were all born at the same time?

Response: Pregnant women were enrolled into the study in rolling fashion over the course of approximately one year, and intervention delivery followed the same rolling order such that interventions were initiated around the time of delivery for each enrolled woman. We have clarified the text as follows.

“Field staff screened the areas to enroll pregnant women **over the course of approximately one year**... Interventions were initiated **in rolling fashion** around when **the enrolled women gave birth**”

4. Procedures line 383/384 – in addition to the average ages, it would be helpful to provide the variability in measurement ages (how long were these cross-sectional surveys?)

Response: Data collection visits followed the same rolling order of intervention delivery, which followed the rolling order of enrollment of pregnant women over the course of one year. Therefore, each measurement timepoint captured a relatively narrow age range. We have added the interquartile range for child age at each measurement timepoint for each country to the results section (under enrollment) as follows.

“In Bangladesh, 5551 pregnant women in 720 clusters were enrolled between 31 May 2012 and 7 July 2013. Antibiotic use was recorded among children in the birth cohort participating in a longitudinal substudy conducted to assess environmental enteric dysfunction (EED), which included 1131 children at mean age 3 months (**interquartile range [IQR]=1.6-4.1 months**), 1531 children at 14 months (**IQR=12.8-15.5 months**) and 1531 children at 28 months (**IQR=27.2-29.7 months**)”

“In Kenya, 8246 pregnant women in 702 clusters were enrolled between 27 November 2012 and 21 May 2014. The EED substudy included 1493 children at mean age 6 months (**IQR=4.1-6.8 months**), 1504 children at 17 months (**IQR=15.3-18.3 months**) and 1444 children at 22 months (**IQR=21.1-23.7 months**)”

5. Procedures outcomes – is the outcome at any of the 3 time points? Or any use over any of the 3 time windows? (specified later in the analysis section, but should be included here)

Response: Each of the three measurement points was treated as a separate outcome datapoint for a given child and the analysis pooled outcome data over the three time points. We have clarified as follows (the pooling over the three datapoints is already described under statistical analysis).

“For the present analysis, our pre-specified primary outcome was the prevalence of children who used antibiotics at least once within 90 days prior to data collection, **tabulated individually at each of the three measurements points.**”

6. Procedures line 422 – “data collection date” is the outcome collection date or something else? Child age is age at outcome data collection? Everything else at baseline? Or at time of outcome ascertainment?

Response: Thank you for pointing this out. We have clarified when the adjustment covariates were measured.

“we considered the following adjustment covariates, **recorded either at the time of outcome ascertainment (date in 3-month intervals, child age and sex) or at the trial’s baseline** (birth order, mother’s age, height and education, household food insecurity, number of individuals <18 years in household, number of individuals living in compound, distance to household’s primary drinking water source, housing materials, and household wealth index calculated from principal components analysis of household assets)”

7. Results lines 129/137 – this is minor, but should just report the prevalence in boys and girls in the text here

Response: We have added the prevalence for boys and girls in the control arm.

“The prevalence of using antibiotics at least once in the last 90 days in the control group was highest (75%) at the 14-month measurement, and similar for boys (**64%**) and girls (**62%**) (Supplementary Table 8).”

“Use appeared highest at the 6-month measurement and similar for boys (**52%**) and girls (**53%**) (Supplementary Table 9).”

8. Results line 153 – report the p value for the heterogeneity test here

Response: We have added the interaction p-values to this statement.

“In Kenya, there was no evidence of effect modification by child age or sex for most comparisons (**interaction p-values>0.20**, Figs 3-6, Supplementary Tables 16, 17).

9. Results line 196 – this interpretation is a little misleading. The effects are very similar in last month/last 2 weeks, but last 2 weeks is not significant, probably because smaller number with outcome (less precision), not because effect is not there

Response: Thank you for making this point. We agree and have revised the text as follows.

“The nutrition and N+WSH interventions also appeared to reduce antibiotic use in the last two weeks by 7-11% but the associations could not be distinguished from chance, which could be due to reduced precision because of the lower prevalence (~20%) of antibiotic use during this shorter window”

10. Because some outcomes are on the absolute and some on the relative scale, additive and multiplicative interaction are both assessed which is great. This is noted in the methods but glossed over in the results – does the evidence for multiplicative and additive interaction always agree? If not, where do they disagree?

Response: The multiplicative and additive effect estimates were primarily in agreement and yielded similar conclusions.

“In both countries, effect modification estimates on multiplicative and additive scales yielded similar conclusions.”

11. Discussion line 215- can you do a mediation analysis to assess whether the reductions observed in diarrhea/resp illnesses (or viral infections, or intervention adherence) are responsible for reductions in antibiotic use?

Response: We fully agree with the reviewer that demonstrating mediation would strengthen our conclusions. The mediation analysis is extensive in scope and is therefore reported in a separate manuscript by another investigator from the trial team. Given the two countries, four study arms and two sets of hypotheses, we believe that two separate manuscripts are necessary to describe and interpret the analyses with adequate clarity and detail. The mediation analysis demonstrated that the reductions in antibiotic use in Bangladesh that we report here were mediated by reductions in caregiver-reported diarrhea, acute respiratory infections with fever, and carriage of enteric viruses in child stool. The mediation analysis is available in preprint. We have added discussion on this point to the text as follows and provided a citation for the preprint, and we also provide the full reference here.

“Additional analysis demonstrated that the reductions in antibiotic use were mediated by reductions in the prevalence of reported diarrhea, reported acute respiratory infection with fever, and detection of enteric viruses in child stool ²⁷.”

27. Nguyen A, Barratt Heitmann G, Mertens A, Ashraf, Rahman MZ, Ali S, Rahman M, Arnold BF, Grembi JA, Lin A, Ercumen A, Benjamin-Chung J. Pathways through which water, sanitation, hygiene, and nutrition interventions reduce antibiotic use in young children: a mediation analysis of a cluster-randomized trial. medRxiv 2024.10.13.24315425; doi: <https://doi.org/10.1101/2024.10.13.24315425>

12. Discussion line 297 – was there agreement between caregiver report and medical report even with a long recall period like 90 days?

Response: This is a very good point. The previous study compared medical records to twice-weekly caregiver report so the observed concordance may not hold over a long recall period. We have added text on this as a limitation.

“A previous study of birth cohorts in eight countries found good agreement between medical reports and caregiver-reported antibiotic use in children **recorded via twice-weekly visits**⁶. **The observed concordance with medical reports may not apply to the longer recall period in our study, and the possibility remains that the reported reductions in antibiotic use in Bangladesh were influenced by courtesy bias or placebo effects.**”

13. Discussion – can the authors comment more on why there were observed differences by sex?

Response: We have added a discussion of possible differences by sex.

“In Bangladesh, interventions reduced the prevalence of using antibiotics at least once in the last 90 days among girls but not boys. This may reflect biological differences, sex-specific behaviors, or differential treatment by caregivers. For example, a recent meta-analysis found that newborn girls exhibited greater growth improvements from prenatal small-quantity lipid-based nutrient supplements compared to boys²⁵. In a birth cohort study in eight countries, girls were slightly less likely to receive antibiotics for diarrheal and respiratory infections than boys⁷. However, we observed similar intervention effects on the prevalence of using antibiotics multiple times, and the total times and days of antibiotic use for girls and boys, suggesting no overall trends by sex.”

Reviewer #1 (Remarks on code availability):

Code and data are not currently publicly available on the provided site. Therefore I was unable to run the code.

Response: The data and code are now available at this link: <https://osf.io/t7fmw/>

Reviewer #2 (Remarks to the Author):

This is a secondary data analysis of WASH Benefits trial data from Bangladesh and Kenya, which mirrors the primary results of the trial – in a setting where there were reductions in diarrhoea (Bangladesh) there were modest reductions in antibiotic use; in a setting where there was no reduction in diarrhoea (Kenya), there was no reduction in antibiotic use.

Antibiotic data were available for a subgroup of children participating in an EED substudy, who had similar characteristics to the broader trial population. My major comment is that there are a huge number of analyses presented, including subgroup

analyses and sensitivity analyses, but these are all underpowered as secondary/tertiary exploratory analyses, and with no adjustment for multiple hypothesis testing.

Response: While antibiotic use data were collected from a subset of children enrolled in the WASH Benefits trial, data were collected from the subset at three timepoints, increasing the total number of observations available for the presented analysis. Additionally, while antibiotic use was not a primary outcome of the parent trial, it had substantially higher prevalence (>50% among controls in both countries) than the primary outcome of the parent trial (diarrhea, 6% among controls in Bangladesh, 27% among controls in Kenya), yielding increased statistical power for hypothesis testing. Therefore, the presented analyses are not underpowered. With the available number of observations, we report 80% power to detect 11% relative reduction in Bangladesh and 13% relative reduction in Kenya in the prevalence of using antibiotics at least once in 90 days. In contrast, the parent trial was powered to detect a 30% relative reduction in diarrhea prevalence [Arnold et al. 2013]. Therefore, the presented analysis is more highly powered than the analyses of the primary outcomes of the trial.

We have not adjusted for multiple hypothesis testing, following the pre-specified analysis plan of the WASH Benefits trial [Arnold et al. 2013, updated on 2016.02.05, <https://osf.io/63mna/>]. Bonferroni corrections and other multiplicity adjustments can be overcorrections, especially if the outcomes are correlated [Schulz and Grimes 2005]. It is possible that this lack of correction increased the risk of Type I error and led to chance findings in our analysis. However, the observed effects are remarkably consistent across the different analyses, and it is unlikely that they would all arise from chance. We have added a discussion of this point to the text.

Arnold BF, Null C, Luby SP, Unicomb L, Stewart CP, Dewey KG, Ahmed T, Ashraf S, Christensen G, Clasen T, Dentz HN. Cluster-randomised controlled trials of individual and combined water, sanitation, hygiene and nutritional interventions in rural Bangladesh and Kenya: the WASH Benefits study design and rationale. *BMJ open*. 2013 Aug 1;3(8):e003476.

Schulz KF, Grimes DA. Multiplicity in randomised trials I: endpoints and treatments. *The Lancet*. 2005 Apr 30;365(9470):1591-5.

“Finally, we did not correct for multiple hypothesis testing because Bonferroni corrections and other multiplicity adjustments can lead to overcorrections⁵¹. Therefore, some of the reported effects could be chance findings, but observed effects were highly consistent across different analyses and unlikely to be explained by chance.”

In particular, the sensitivity analyses show that the inferences change if the recall period for antibiotics is 2 weeks, rather than 1 month of 90 days, but one would expect that recall of antibiotic use is actually most reliable in the prior 2 weeks, and much less so in the 30-90 days prior.

Response: We agree that 2-week recall is likely to be more accurate than longer recall windows, but the shorter window also corresponds to lower prevalence of antibiotic use, reducing statistical power for analysis. We note that the magnitude of point estimates for the 2-week recall window is comparable to the other recall windows, but the

associations cannot be distinguished from chance, reflecting reduced precision due to the less frequent outcome measure. We have added a discussion of this point to the text. Please also see our response to comment #9 from Reviewer 1 above.

“The nutrition and N+WSH interventions also appeared to reduce antibiotic use in the last two weeks by 7-11% but the associations could not be distinguished from chance, which could be due to reduced precision because of the lower prevalence (~20%) of antibiotic use during this shorter window”

A few other comments:

1. Were antibiotics used for prophylaxis excluded? For example, in Kenya where HIV-exposed children were likely prescribed cotrimoxazole throughout the period of breastfeeding?

Response: We have added text that we did not collect this information.

“We did not record whether the antibiotic was taken therapeutically or prophylactically, whether it was prescribed and where it was obtained.”

2. Where were antibiotics obtained from? Did the trial provide treatment of illness?

Response: The trial did not provide any medication or treatment of illness, except for deworming medication provided after all outcome data collection was completed. We have added this information to the text. We have also clarified that we did not record where the antibiotic was obtained (see excerpt under comment #1 above).

“The promoters did not provide any medical treatment or advice.”

3. Contacts between the study team and the households were not the same in the intervention and control arms. Did fieldworkers provide advice about management of illness during their visits to active arms, and thereby influence behaviours of caregivers potentially? In which case, this reduction would be more due to behaviour change communication, rather than WASH per se.

Response: We have clarified that the promoters did not provide advice about management of disease (see excerpt under comment #2 above).

The reviewer raises an important point about the role of contact with the study team. In Bangladesh, the study team engaged more frequently with households in intervention arms because the community health promoters did not visit households in the control arm; therefore, there is a possibility that interactions with the study team influenced participant behaviors in addition to the intentional promotion of the interventions. In Kenya, the trial included a passive control arm who received no visits from community health promoters, as well as an active control arm where community health promoters visited monthly to measure child mid-upper arm circumference but did not provide any

information on water, sanitation, hygiene, nutrition or management of disease. The current analysis uses data from the active control arm. The active control arm was included in the Kenya trial to isolate the influence of engagement with the study team from the influence of the interventions. The primary outcomes of the trial (diarrhea prevalence, length-for-age z-scores) were indistinguishable between the active and passive control arms, indicating no standalone effect from engagement with the study team in the absence of interventions. We have added this information as follows:

“In Kenya, the trial included both passive and active control arms to isolate the influence of engagement with the study team from intervention effects. Promoters did not visit passive control households. They visited active control households monthly to measure child mid-upper arm circumference **but did not promote any behavior change.**”

The EED substudy was conducted in nutrition, WSH, N+WSH and control arms (**passive in Bangladesh, active in Kenya**) of the parent trial.

“Another possible explanation could be that the control arm in Bangladesh was passive (not visited by community health promoters) while the EED substudy in Kenya drew from the active control arm where community health promoters visited households regularly but did not promote any behaviors. Therefore, in Bangladesh, it is possible that additional interaction with the study team among intervention households influenced participant behaviors in addition to the interventions, potentially leading to stronger effects compared to passive controls. However, in Kenya, the primary trial outcomes were indistinguishable between the active and passive control arms²², indicating no standalone effect from interaction with promoters.”

4. The major effects seen in Bangladesh were at 3 months of age – what did the nutrition intervention comprise at this age, and what about WASH at this young age?

Response: The effects were strongest at the 3-month measurement. We have added information on how the different interventions may have affected this age group.

“For this age group, the nutrition intervention included recommendations for maternal dietary diversity from pregnancy through lactation, early initiation of breastfeeding and exclusive breastfeeding until 6 months, while effects of water, sanitation and hygiene interventions may be mediated through cleaner caregiver hands and a more hygienic domestic environment. Notably, in this age group, all three interventions were associated with a 28-44% reduction in caregiver-reported acute respiratory infections among children enrolled in the EED subset.”

5. Some statements need revising to be more nuanced: eg “immune systems weakened by poor nutritional status put children at risk of further infections”; and “Repeated antibiotic use within a short period persistently alters the gut microbiota into a predominantly resistant population”.

Response: We have expanded these statements as follows.

“Repeated antibiotic use within a short period persistently alters the gut microbiota into a predominantly resistant population **by exerting selective pressure in favor of resistant strains**”³¹.

“Repeated episodes of diarrhea can lead to malnutrition, **and malnourished children in turn experience increased incidence, duration and severity of diarrhea (“infection-malnutrition cycle”)**”¹⁵.

Reviewer #3 (Remarks to the Author):

***What are the noteworthy results?

This study reanalyzes data from the WASH Benefits trial that measured the impact of improvement in water, sanitation, and hygiene (WSH), nutrition (N), or both (WSH+N) on a variety of environmental, behavioral, and clinical health outcomes. In this case, the study examines the impact of the trial on caregiver-reported antibiotic usage in children in both Bangladesh and Kenya sites overall and by age group. Noteworthy results include demonstration of high levels of reported antibiotic use for multiple types of antibiotic classes between birth and 28 months of age and demonstration of similar country-level patterns of impact of WSH, N, and WSH+N interventions on reported antibiotic use as observed in analysis of trial impact on self-reported diarrhea and respiratory symptoms. This study also finds interaction between trial reductions in usage by age with strongest impact on the youngest ages.

***Will the work be of significance to the field and related fields? How does it compare to the established literature? If the work is not original, please provide relevant references.

While studies on antibiotic use are common, evidence on how WSH and nutrition interventions impact antibiotic use are relatively few in number and WASH Benefits is considered a powerful study for examining causal relationships between WSH and nutrition conditions and early childhood health. This study will be well received by the science community.

***Does the work support the conclusions and claims, or is additional evidence needed? Are there any flaws in the data analysis, interpretation and conclusions? Do these prohibit publication or require revision?

Part of the work supports the conclusions and claims although the discussion lacks some critical points about study design factors that could explain the inconsistent study results between Bangladesh and Kenya and thus the need for caution in inference about generalizability of WSH+N impacts on antibiotic usage across different settings.

Response: Thanks for raising this point. Please see our responses under individual comments below.

Recommended edits prior to publication are as follows:

MAJOR COMMENTS

1. Authors address the limitation of using self/caregiver-reported antibiotic usage in the discussion. However, the methods suggest that they may have an observable outcome that could be compared to reported outcomes as an assessment of potential reporting bias. Specifically, the methods state that if participants reported antibiotic usage, then field staff asked to see the prescription or packaging for the prescription. At minimum, the proportion of caregivers in each group in each site who produced an antibiotic or package for inspection should be reported in Table 1 and in the Results paragraph on usage. If sufficient data is available, it would also be valuable as an outcome in models. This would allow for a more nuanced discussion on reliability of the reported antibiotic usage outcome in the discussion, and potentially even bolster the self-reported outcome results.

Response: While data collectors were instructed to check prescriptions or packaging when available, they did not record information on which caregivers were able to produce these for validation. We have clarified this point in the text as follows.

“We did not record whether the caregiver was able to produce the prescription or packaging for the antibiotic.”

Therefore, the limitation remains that the outcome data in the presented analysis is caregiver-reported and potentially subject to bias. However, in a new analysis led by our team that used pathogens detected in child stool samples, the observed reductions in caregiver-reported antibiotic use were mediated by reductions in objectively measured virus detection in stool (as well as by reductions in caregiver-reported diarrhea and acute respiratory infections with fever), lending support to a causal interpretation that the interventions reduced antibiotic use by reducing infections, and they particularly reduced unnecessary antibiotic use for viral infections. This mediation analysis is available in preprint. We have added discussion on this point to the text as follows and provided a citation for the preprint, and we also provide the full reference here. Please also see our response to comment #11 from Reviewer 1 above.

“Additionally, at the 14-month measurement point in the EED cohort in Bangladesh, TaqMan Array Card analysis of stool samples found that children in the WSH, nutrition and N+WSH groups carried fewer viruses and children in the WSH group carried fewer total pathogens than controls but there was no effect on carriage of bacterial pathogens in any intervention group ²⁶. We found lower antibiotic use in all three of these study arms. **Additional analysis demonstrated that the reductions in antibiotic use were mediated by reductions in the prevalence of reported diarrhea, reported acute respiratory infection with fever, and detection of enteric viruses in child stool ²⁷. Mediation through reduced respiratory infections (often of viral etiology) and carriage of enteric viruses suggests that interventions may have reduced uncalled-for antibiotic prescriptions prompted by viral infections, consistent with previous evidence that antibiotics are often unnecessarily prescribed for**

child diarrhea and respiratory infections in LMICs^{7,8}.”

27. Nguyen A, Barratt Heitmann G, Mertens A, Ashraf, Rahman MZ, Ali S, Rahman M, Arnold BF, Grembi JA, Lin A, Ercumen A, Benjamin-Chung J. Pathways through which water, sanitation, hygiene, and nutrition interventions reduce antibiotic use in young children: a mediation analysis of a cluster-randomized trial. medRxiv 2024.10.13.24315425; doi: <https://doi.org/10.1101/2024.10.13.24315425>

2. Page 10 includes a thorough proposed explanation for different effects of the WSH, N, and WSH+N intervention in Bangladesh versus Kenya. Another possible explanation that needs to be acknowledged and discussed after this paragraph is the potential for differential courtesy bias caused by study design. The passive control approach in Bangladesh vs active control approach in Kenya meant controls had different levels of contact with the study team. Kenyans with greater contact with the study team may have over or under reported relative to Bangladeshi participants based upon knowledge and reaction to being closely studied - such as altering responses based upon beliefs about what responses might be expected or desired of them. Similarly, Bangladeshi trial participants with promoters visiting six times per month may have over or under-reported usage based upon knowledge of trial status and perception that certain types of responses may be expected of them relative to trial participants in Kenya. Alternatively, the observed differences in usage as well as self-reported diarrhea in Bangladesh may be factual but be more due to the impact of health promoters visiting participants so frequently and less about WSH and N intervention components. The fact that WASH-B impacted self-reported diarrhea and respiratory symptoms (and antibiotic usage) to a much greater extent than anything observed with objectively measured pathogen infections or environmental indicators in Bangladesh is concerning, given that the impact of the trial on reported and objective measures in Kenya was pretty consistent. Thus, study design should be acknowledged as an potential influence on trial effects, at least in the context of this study on reported antibiotic usage.

Response: The reviewer raises an important point about the role of engagement with the study team. Another reviewer brought up a similar point – please see our response to comment #3 from Reviewer 2 above.

Notably, the reductions observed in objectively measured outcomes (pathogen detection in child stool) were on par in magnitude with the reductions observed in the self-reported outcomes. These include 17-33% reduction in the prevalence of parasite carriage and 42-65% reduction in the prevalence of virus carriage in stool. These findings lend strength to a causal interpretation of intervention effects on self-reported outcomes. We have added this information to the discussion section as follows.

“Our findings are also consistent with reductions in objectively measured outcomes in the Bangladesh trial. Children receiving WSH and N+WSH interventions had 17-25% lower carriage of *Giardia*⁴⁹ and 29-33% lower carriage of hookworm⁵⁰ in stool, compared to controls. Children receiving the WSH intervention had 49-65% lower carriage of enteric viruses (norovirus, sapovirus and adenovirus) and those receiving the nutrition intervention had 42% lower carriage of sapovirus²⁶. Further, the reductions in virus carriage were found to mediate the observed

reductions in antibiotic use²⁷. Taken together, these findings lend support to a causal interpretation of intervention effects on caregiver-reported antibiotic use.”

3. If your hypothesis about WSH+N impact on antibiotic usage being an outcome of WSH+N reductions in diarrhea/respiratory symptoms is correct, then in the discussion about why Kenya did not experience the same WSH+N impact on antibiotic usage, you could acknowledge that no reduction in usage would be expected if primary sources of infections in Kenya are from sources that are poorly addressed by the WSH+N design, for example food sources, flies, or animals.

Response: This is a good point. We have added text as follows and provided references to support this point.

“The lack of intervention effects on antibiotic use in Kenya could also indicate that the household-level WSH and nutrition interventions did not address dominant drivers of infectious disease transmission in this setting. These could include contaminated food sources, domestic animals, and poor community-wide sanitation^{29,30}.”

4. In the discussion of limitations, please add that this study did not document the motivations for antibiotic usage so if usage is more motivated by beliefs and factors external to WSH or nutrition (e.g. use for other symptoms, beliefs that antibiotics enhance breastmilk quality or growth and development of the infant) then there should be no effect of WSH or N interventions on antibiotic use, even if WSH and N reduce diarrhea or respiratory symptom prevalence. The relationship between drivers and usage could also explain the differences observed between countries in this study if usage in Bangladesh is motivated by diarrhea or respiratory symptoms, by WSH and/or N knowledge, or by courtesy bias but in Kenya usage is motivated more by the factors cited above.

Response: Thank you for this nuanced point. We have added to the text as follows:

“Alternatively, if antibiotic use is motivated by treating diarrhea and respiratory infections in Bangladesh but by beliefs and preferences external to WSH and nutrition (e.g., to enhance child growth, treat other symptoms) in Kenya, differences in findings between the two countries may also be explained by differences in drivers of antibiotic use.”

Minor comments

1. Several places in the manuscript read as if antibiotic usage was observed or validated, even if that was unintended. More consistent use of "reported antibiotic usage" or "caregiver-reported antibiotic usage" at key points such as in tables and figures, in the Introduction final paragraph, and at the beginning of the discussion is needed to ensure readers understand the nature of the outcome measure.

Response: We agree with the reviewer. We have replaced “antibiotic use” with “caregiver-reported antibiotic use” in all tables and figure captions, in the final paragraph of the introduction and first paragraph of the discussion as the reviewer suggested, as

well as at first mention in the abstract and methods.

2. The last sentence of the first paragraph on page 3 is not correct because it has flipped the results of these two papers. References 6 and 7 measured antibiotic use in people with diarrhea and respiratory symptoms, not the primary reasons for antibiotic use. Reasons for antibiotic use are also commonly unrelated to diarrhea and respiratory symptoms, such as prescriptions for general fevers, rashes, or other symptoms, as well as self-medication choices based upon beliefs that antibiotics enhance physical growth, ensure healthy pregnancies and the quality of breastmilk, among other cultural beliefs.

Response: We have corrected this sentence as follows.

“Antibiotics are commonly prescribed in LMICs for diarrheal and respiratory infections^{7,8}. While cultural factors such as the beliefs and perceptions of both the prescribers and consumers can drive antibiotic use¹⁰, reducing the occurrence of diarrheal and respiratory infections may lead to reduced antibiotic use in LMICs.”

3. On page 2 and 11, language references primary drivers of antimicrobial resistant emergence and transmission but does not acknowledge the importance of zoonotic sources. Many consider antibiotic use in animals a more important driver of AMR transmission and carriage than human sources so acknowledging zoonotic vectors in these two locations would be more holistic.

Response: This is a great point. We have added text on antibiotic use by animals and zoonotic transmission of AMR to the introduction and discussion sections as follows and supported these additions with references.

“Reasons may include densely populated conditions, lack of safe drinking water and sanitation⁴, and widespread availability and frequent use of antibiotics **for both humans and animals**^{5,6}.”

“Notably, antibiotic use for domestic animals and zoonotic transmission of antimicrobial resistant organisms are important contributors to community carriage of AMR in LMICs where humans and animals often share living spaces and animal fecal waste is not safely managed³⁹.”

6. Van Boeckel, T. P. *et al.* Global trends in antimicrobial resistance in animals in low- and middle-income countries. *Science* 365, eaaw1944 (2019).

39. Swarthout, J. M., Chan, E. M. G., Garcia, D., Nadimpalli, M. L. & Pickering, A. J. Human Colonization with Antibiotic-Resistant Bacteria from Nonoccupational Exposure to Domesticated Animals in Low- and Middle-Income Countries: A Critical Review. *Environ. Sci. Technol.* 56, 14875–14890 (2022).

4. The end-of-paragraph sentence at the beginning of page 10 about viral vs bacterial differences in WASH-B trial impact should be followed by the subsequent observation that if trial arms did reduce antibiotic usage because of reduction in symptomatic viral infections, that lower antibiotic usage did not reduce or increase the probability of trial or control participants experiencing a bacterial infection.

Response: Children in low-income countries are often asymptotically colonized by pathogens. We measured carriage of bacterial pathogens in child stool but did not assess whether carriage resulted in symptomatic diarrhea. Antibiotic use has been associated with increased child diarrhea, and conversely, reducing unnecessary antibiotic use has been estimated to reduce child diarrhea [Rogawski et al. 2015; 2027]. The interventions that reduced antibiotic use in our analysis also reduced symptomatic diarrhea in the trial cohort. Therefore, it is possible that the reductions in unnecessary antibiotic via reduced viral infections did reduce diarrhea among children despite having no effects on gut carriage of bacterial pathogens.

Rogawski ET, Westreich D, Becker-Dreps S, Adair LS, Sandler RS, Sarkar R, Kattula D, Ward HD, Meshnick SR, Kang G. The effect of early life antibiotic exposures on diarrheal rates among young children in Vellore, India. *The Pediatric infectious disease journal*. 2015 Jun 1;34(6):583-8.

Rogawski ET, Meshnick SR, Becker-Dreps S, Adair LS, Sandler RS, Sarkar R, Kattula D, Ward HD, Kang G, Westreich DJ. Reduction in diarrhoeal rates through interventions that prevent unnecessary antibiotic exposure early in life in an observational birth cohort. *J Epidemiol Community Health*. 2016 May 1;70(5):500-5.

5. Page 13, please add the usage recall period used in reference 6 to the statement "However, a previous study of birth cohorts in eight countries found good agreement between caregiver-reported antibiotic use in children and medical reports 6."

Response: Thanks for raising this point. The previous study compared medical records to twice-weekly caregiver report so the observed concordance may not hold over a long recall period. We have added text on this as a limitation.

“A previous study of birth cohorts in eight countries found good agreement between medical reports and caregiver-reported antibiotic use in children recorded **via twice-weekly visits**⁶. **The observed concordance with medical records may not apply to the longer recall period in our study, and the possibility remains that the reported reductions in antibiotic use in Bangladesh were influenced by courtesy bias or placebo effects.**”

***Is the methodology sound? Does the work meet the expected standards in your field?

The approach to measurement of antibiotic use is consistent with standards in the field. Authors address the challenges in accurate measurement of usage and propose rigorous ways to improve that measurement in the future. The design of the WASH-B parent trial that was leveraged for this sub-study is rigorous. The analytical approach is standard but suited to the research hypothesis.

***Is there enough detail provided in the methods for the work to be reproduced?

The methods provide enough detail for the work to be reproduced, especially in combination with prior publications from the same authors on study design details.

Reviewer #3 (Remarks on code availability):

I have skimmed the code and it appears usable although i have not replicated the results due to lack of time to do so with rigor.